# Acetamiprid-Induced Toxicity Thresholds and Population Sensitivity in *Trichogramma dendrolimi*: Implications for Pesticide Risk Assessment

**DOI:** 10.3390/insects16070698

**Published:** 2025-07-07

**Authors:** Yan Zhang, Jiameng Ren, Shenhang Cheng

**Affiliations:** 1Department of Biological Science and Technology, Jinzhong University, Jinzhong 030619, China; sherrytjnu@163.com (Y.Z.); renjiameng08@aliyun.com (J.R.); 2School of Life Science, Shanxi University, Taiyuan 030006, China; 3School of Synthetic Biology, Shanxi University, Taiyuan 030006, China

**Keywords:** ecotoxicology, acetamiprid, risk assessment, sublethal effects, species sensitivity distribution

## Abstract

Insects like the *Trichogramma dendrolimi* are essential for controlling harmful pests in agriculture. However, pesticides used to kill these pests can also harm beneficial insects like *T. dendrolimi*. This study examined the effects of a common pesticide, acetamiprid, on *T. dendrolimi*. It found that even small amounts of acetamiprid can reduce the ability of *T. dendrolimi* to lay eggs and survive, without necessarily killing them. By using a statistical model, the study also showed that acetamiprid could affect a large portion of similar beneficial insects at commonly used field concentrations. These findings highlight the importance of considering not just whether a pesticide kills insects, but also its subtle effects on beneficial insects that help control pests naturally. This information can help improve pest management strategies to better protect both crops and beneficial insects.

## 1. Introduction

Integrated pest management (IPM) strategies have evolved to emphasize the synergistic use of biological and chemical controls, aiming to maximize pest suppression while minimizing environmental impacts [1]. Chemical pesticides continue to play a pivotal role in pest management, yet their application often poses risks to beneficial organisms, such as parasitic natural enemies [2,3,4,5]. This is particularly true for neonicotinoid insecticides like acetamiprid, which are widely used in agriculture but have been implicated in disrupting the normal physiological functions of insects, including parasitoids. Parasitoid wasps of *Trichogramma* (Hymenoptera: Trichogrammatidae) are keystone biocontrol agents, with over 219 recognized species [6,7,8]. Their application in pest control has emerged as a cornerstone of integrated pest management (IPM) strategies across numerous countries [7,9,10,11,12]. In northeastern China, *T. dendrolimi* dominates corn ecosystems, comprising 45–97% of natural parasitoid communities [13], and is extensively used against *Conogethes punctiferalis* [14,15]. However, the efficacy of these beneficial insects is threatened by the widespread use of chemical pesticides, particularly neonicotinoids like acetamiprid, which can affect non-target organisms, including parasitoids [16,17,18].

Neonicotinoids, acting as nicotinic acetylcholine receptor (nAChR) antagonists, have been implicated in disrupting the normal physiological functions of insects, including parasitoids [19,20]. Prior studies have quantified the acute toxicity of neonicotinoids to various *Trichogramma* species [3,21], yet the sublethal impacts and population-level sensitivity variations remain understudied. Recent research has shed light on the compatibility of novel pesticides with parasitoids, demonstrating varying degrees of toxicity [22,23].

To fully understand the ecological risks associated with pesticide use, it is crucial to evaluate both the acute and sublethal effects on non-target organisms. Sublethal doses of pesticides can induce physiological and behavioral changes that may significantly impact parasitism rates, developmental periods, fecundity, and offspring emergence [24,25,26,27]. These sublethal effects are often overlooked in traditional toxicity assessments but can have profound implications for biological control services. Therefore, a comprehensive assessment of pesticide impacts on parasitoids is essential for developing effective and ecologically sound pest management strategies.

Advances in the assessment of pesticide impacts on ecosystems in China have been integrated into comprehensive prevention strategies, driven by a deeper understanding of chemical pesticide toxicity and the establishment of standardized protocols, such as China’s Guidelines on Environmental Safety Assessment for Chemical Pesticides (Standard No. HJ/T 88-2016) [28]. However, there remains a gap in identifying indicator species among parasitic natural enemies within these guidelines. Species sensitivity distributions (SSDs) have emerged as a valuable tool for estimating the sensitivity of aquatic organisms to stressors and have the potential to be applied more broadly in terrestrial ecological scenarios [29,30,31,32,33].

The primary objectives of the present study were as follows: (1) to evaluate the acute toxicity of acetamiprid residues to adult *T. dendrolimi*, including determining the lethal concentration (LC_50_) and associated toxicity, (2) to investigate the long-term sublethal effects (here defined as continuous exposure spanning at least one full generation or 10–14 days, covering critical life stages) of acetamiprid exposure on key biological traits of *T. dendrolimi*, including parasitism percentage, emergence percentage, offspring survival rate, development period of surviving female parasitoids and identify the no-observed-effect concentrations (NOEC) for acetamiprid in *T. dendrolimi* under sublethal exposure conditions, (3) to assess the sensitivity distribution of *T. dendrolimi* among parasitic natural enemies under acetamiprid stress using species sensitivity distribution (SSD) models, and (4) to compare the relative sensitivity of *T. dendrolimi* with other parasitoid species. This study provides critical insights into the ecological risks of acetamiprid to *T. dendrolimi* and its role as a potential bio-indicator for assessing pesticide impacts on parasitoid communities. The findings help to develop pest management strategies that are more effective and ecologically sound.

## 2. Materials and Methods

### 2.1. Insects

*T. dendrolimi* colonies were maintained at the Beijing Academy of Agricultural and Forestry Science (BAAFS). *Corcyra cephalonica* Stainton (Lepidoptera: Pyralidae) eggs were used as the substrate for colony rearing. Prior to use, the eggs were subjected to UV sterilization under a UV lamp emitting at 254 nm for 30 min to ensure no contamination. Subsequently, the disinfected eggs were glued onto cardboard strips (dimensions. 1.5 cm × 6 cm) using an adhesive agent. These eggs were then exposed to newly emerged (24–48 h old) *T. dendrolimi* females in glass tubes (dimensions: 9.5 height × 2.2 cm diameter, covered with black cloth) for a period of 24 h to ensure near-complete parasitism (approximately 100%).

After 24 h of exposure, the cardboard strips were transferred to new containers. A cotton thread soaked in a 10–15% honey solution was provided in the glass tubes to serve as a food source for the newly emerged wasps. All parasitoids were maintained collectively in controlled conditions (25 ± 1 °C, 60 ± 10% RH, and 14:10 photoperiod) in an incubator. Newly emerged females (24–48 h) were used for bioassays.

### 2.2. Pesticides

For this investigation, two insecticides were identified as active candidates: acetamiprid (97% purity, Qingdao Hailir Pesticides and Chemicals Group Co., Ltd., Qingdao, China) serving as the experimental agent and dimethoate (98% purity, Qingdao Hailir Pesticides and Chemicals Group Co., Ltd.) designated as the reference standard. Additionally, they represent distinct chemical classes, providing a comprehensive perspective on the behavior of different pesticide types. The active ingredients of the pesticides were used rather than their commercial formulations to eliminate potential interference from other components in the formulations. Due to their low solubility in water, the stock solutions were prepared by dissolving the pesticides in analytical-grade acetone before each experiment. Seven doses of acetamiprid, ranging from a low concentration to the highest dose (water, 0, 0.23, 1.15, 2.3, 11.5, 57.5 μg a.i. L^−1^) in a geometric series, were carefully measured and prepared in the stock solutions. These doses approximated the intended concentrations, ensuring accuracy.

### 2.3. Acute Contact Toxicity Testing

The experimental design was structured to assess the effects of pesticide residues on adult parasitoid wasps (*T. dendrolimi*) in a dose-dependent manner. Glass-vial residue assays were conducted with seven acetamiprid concentrations (geometric series) applied to a tube (height: 8.0 × 2.5 cm). Dimethoate was included solely as a positive control (validating assay sensitivity via high mortality at a fixed concentration. For solution application, 1 mL of each pesticide solution was uniformly applied to ensure complete coverage of the internal glass surface. A solvent control group used 1 mL of pure acetone, while a blank control group used the same of deionized water. The tubes were rotated with a pipe roller until no droplets remained on the glass wall and were left at room temperature for 1 h to ensure complete evaporation of acetone. For parasitoid exposure, 100 ± 10 adult parasitoids were placed in each tube, which contained a cotton wool plug soaked with a 10–15% honey solution. The tubes were covered with fine gauze to allow air circulation and maintained under controlled environmental conditions. Mortality assessment was conducted after 24 h of exposure, with the number of dead wasps counted based on the absence of movement. Three independent replicates were conducted for each concentration of each compound, and data were recorded and analyzed to calculate mortality rates. This experimental design ensured precise control of exposure conditions while maintaining consistency across treatments, providing robust data for assessing pesticide effects on adult parasitoids.

### 2.4. Effects of Sublethal Treatment of Pesticides on the Development of T. dendrolimi

The sublethal toxicity test was conducted using a modified version of the acute contact toxicity test protocol, with adjustments in experimental design and measurement endpoints to assess the sublethal effects of acetamiprid on the development and reproductive success of *T. dendrolimi*. The experiment utilized five sublethal application rates of acetamiprid, determined as fractions (1/2, 1/5, 1/10, 1/50, and 1/100) of the acute 24 h LC_50_ value obtained from a previously described acute contact toxicity test (method adapted from Yu et al., 2014 who tested imidacloprid) [34]. These fractions were chosen to encompass a broad range of sublethal effects, allowing for a comprehensive evaluation of the impact of acetamiprid on *T. dendrolimi*. Solvent and blank controls were included to ensure accurate baseline measurements. Each treatment group consisted of 50 ± 10 adult wasps, with five independent replicates per treatment. LC_50_ values were calculated using probit analysis with 95% confidence intervals. Data were presented following OECD Guideline 245 format. The NOEC was determined as the highest concentration demonstrating no statistically significant difference from controls in survival/reproduction endpoints. Adult wasps were exposed to the respective acetamiprid concentrations for one hour in glass tubes under controlled conditions (25 ± 1 °C, 70 ± 10% RH, 14:10 L:D photoperiod). After the exposure period, a cardstock strip (1 × 7 cm) containing 300 freshly laid, untreated host eggs (*Corcyra cephalonica* Stainton) was introduced into each glass tube to allow parasitism. The eggs were not irradiated or cold-stored before being used in the experiment.

Following a three-day exposure period, the parasitized eggs were transferred into clean, pesticide-free glass tubes. All experimental units were maintained in climate-controlled incubators under the same conditions as previously described. The number of successfully parasitized eggs was counted daily, and these eggs were deemed successfully parasitized if they exhibited a black and plump appearance. It is important to note that due to the potential for superparasitism in *T. dendrolimi*, where multiple wasps may parasitize the same egg, the number of parasitized eggs observed does not necessarily equal the number of emerging adult wasps. To account for this, the percentage of parasitized eggs that successfully developed into adult wasps was calculated based on the number of emerging adults observed in each replicate, rather than the initial count of parasitized eggs. Newly emerged adult wasps from each replicate (typically cohorts of 10–20 individuals) were collectively transferred into clean, chemical-free glass tubes (2.5 × 15 cm) containing fresh honey solution (10–15%). This group transfer approach maintained treatment-specific cohorts while minimizing handling stress, with all tubes replaced daily to ensure hygiene. Survival rates were monitored daily over two weeks. The duration from egg parasitism to adult emergence was recorded for each replicate.

### 2.5. Species Sensitivity Distribution

SSDs were established to evaluate the acute toxicity biological parameters of acetamiprid against *T. dendrolimi* and other eleven parasitoid wasps, including *Peristenus spretus*, *Trichogramma ostriniae*, *Trichogramma confusum*, *Encarsia formosa*, *Ganaspis brasiliensis* (Ihering), *Ganaspis* cf. *Brasiliensis*, *Trichogramma nubilale*, *Trichogramma evanescens*, *Trichogramma japonicum*, *Cotesia flavipes*, providing a comparative framework for assessing species sensitivity. The acute toxicity data were compiled from multiple sources, including the US-EPA ECOTOX database (https://cfpub.epa.gov/ecotox/search.cfm), peer-reviewed literature, and draft assessment reports of acetamiprid. Recently published studies [35,36,37] were also included to ensure the analysis captures the latest advancements. The analysis assumed a log-normal distribution of the toxicity data, which was verified using the Anderson–Darling test within the ETX software package, version 2.0 [38]. This statistical approach allowed for the 5th percentile (HC_5_) and 50th percentile (HC_50_), along with their confidence limits, following the methodology outlined by Aldenberg and Jaworska (2000) [39]. The significance level for the Anderson–Darling test was set at 5%. PAF values were derived from the SSD curves by interpolating the fraction of species expected to be affected at a given concentration. The SSD curves were constructed using the log-logistic distribution model, which provided the HC_5_ and HC_50_ values. PAF values at the field-recommended concentration range (30–100 mg a.i. L^−1^) were then calculated by determining the fraction of species with acute toxicity endpoints below these concentrations. This calculation allowed us to estimate the potential impact of acetamiprid on parasitoid communities under field conditions. This approach provided a robust framework for evaluating species sensitivity to acetamiprid, enabling the identification of critical thresholds for ecological risk assessment.

### 2.6. Statistical Analysis

The corrected mortality of *T. dendrolimi* was calculated using the Abbott (1925) [40] formula. The 24 h LC_50_ values and their 95% confidence intervals for the insecticide were determined through probit analysis using SPSS 21.0 (IBM Corp., New York, NY, USA). Data normality and homogeneity of variances were verified by Shapiro–Wilk test and Levene’s test, respectively. Sublethal effect data (parasitism, emergence rate, etc.) were analyzed by one-way analysis of variance (ANOVA) followed by Tukey’s HSD post hoc test for multiple comparisons (*p* < 0.05). All percentage data were arcsine square-root transformed prior to analysis to meet ANOVA assumptions. Results are presented as mean ± standard error (SE). The NOEC was calculated via Dunnett’s multiple comparisons test. Data from the sublethal effect experiments were analyzed using one-way analysis of variance (ANOVA) to assess statistical significance (*p* < 0.05). Significant differences between treatment groups were identified using the Tukey–Kramer honestly significant difference test at *p* < 0.05 level.

## 3. Results

### 3.1. Acute Contact Toxicity and Risk Assessment

In the acute toxicity tests, LC_50_ values for acetamiprid in active ingredient were calculated separately for each of the eight tests performed. The results are presented as box plots in Figure 1. The complete statistical parameters for all toxicity tests (LC50 values with 95% CIs, slopes ± SE, and χ^2^ values) are provided in Appendix A. The mortality in both solvent and blank control did not exceed 15%. The LC_50_ values of acetamiprid for *T. dendrolimi* were 0.80, 0.12, and 0.06 mg a.i. L^−1^ after different exposure times. The results also indicate that the endpoint of toxicity decreased with the increase in exposure time. The risk assessment results showed that acetamiprid was slight to moderately toxic to *T. dendrolimi*.

The dimethoate constantly produces 40–90% cumulative mean adult mortality at the concentration of 0.27 mg a.i. L^−1^ (Figure 2), establishing baseline sensitivity thresholds. Time-dependent assays were not conducted for dimethoate, as its role was to validate assay sensitivity at 24 h (standard duration for reference insecticides). The histogram data is based on 8 experiments with different *T. dendrolimi* strains. These results confirm methodological consistency with established organophosphate toxicity profiles for parasitoids.

### 3.2. The Effect of Acetamiprid on the Development of T. dendrolimi

The sublethal effects of a single application of acetamiprid on parasitism, emergence rate, developmental period, and survival were evaluated. Significant differences were identified by Tukey’s HSD test (*p* < 0.05). No significant differences were observed between the solvent control and blank control groups. However, parasitism rate significantly decreased at 11.5 and 57.5 μg a.i. L^−1^ (F_5,24_ = 8.41; *p* < 0.001). Similarly, emergence rate showed similar trends (F_5,24_ = 7.92; *p* < 0.001) (Figure 3A). Additionally, the Developmental period was reduced at 57.5 μg a.i. L^−1^ (F_5,24_ = 5.63; *p* < 0.001) (Figure 3B). In contrast, none of the tested concentrations had a significant effect on survival. The lowest observed effect concentration (LOEC) for parasitism and emergence rate was determined to be 11.5 μg a.i. L^−1^, while the no-observed-effect concentration (NOEC) was established at 2.3 μg a.i. L^−1^.

### 3.3. Comparison of the Sensitivity to Acetamiprid Between T. dendrolimi and Other Parasitoid Wasps

The HC_5_ and HC_50_ values (0.11 mg a.i. L^−1^ and 5.88 mg a.i. L^−1^ ) with their corresponding 95% confidence interval, obtained from a log-logistic distribution model based on the median lethal concentration (LC_50_) of twelve parasitoids to acetamiprid, are presented in Table 1. The goodness of fit of toxic data was accepted by the Kolmogorov–Smirnov test (*p* = 0.895), Cramer von Mises test (*p* = 0.126), and Anderson–Darling test (*p* = 0.752), at the 5% significance level for the SSD curves. Figure 4A shows the probability density diagram and histogram of acute toxicity data of acetamiprid to parasitoids after logarithmic transformation. The logarithmic value of the acute toxicity data of the acetamiprid presents a bell-shaped distribution, which accords with the logarithmic-logistic distribution model. Figure 4B is the Q-Q diagrams of logarithm values of acute toxicity data. It can also be seen intuitively from the diagrams that the logarithm values of acute toxicity data are all near the theoretical distribution straight line, indicating that they conform to log-logistic distribution. The results show that the log-logistic distribution model can well reflect the sensitivity distribution of parasitoids to acetamiprid.

In addition, according to the maximum exposure concentration (100 mg a.i. L^−1^), the potentially affected fraction (PAF) of species exposed to acetamiprid was 76.8%, which means that at this concentration, 76.8% of the parasitoids will be affected and the result indicated a certain risk to the ecosystem. By analyzing the SSD curves constructed for acetamiprid (Figure 5), the *Peristenus spretus* [41] appears to be among the most sensitive parasitic species for which toxicity data were available, followed by *Trichogramma ostriniae* [42], *T. dendrolimi* (this paper), *Trichogramma confusum* [42], *Encarsia formosa* [43], *Ganaspis brasiliensis* [36], *Ganaspis* cf. *Brasiliensis* [37], *T. nubilale* [44], *T. evanescens* [44], *T. japonicum* [45], *T. ostriniae* [46], *T. confusum* [47] and *Cotesia flavipes* [35].

## 4. Discussion

The documented toxicity of neonicotinoids to *Trichogramma* parasitoids has been well established through multiple studies, including acetamiprid’s effects on *T. chilonis* and *T. brasiliensis* [48] and thiamethoxam’s toxicity to *T. platneri* [49] and *T. chilonis* [50]. Our research significantly expands this understanding by conducting a systematic evaluation of both acute toxicity and sublethal impacts of acetamiprid on *T. dendrolimi*, a crucial biocontrol agent in northeastern China’s agricultural ecosystems. Of particular concern, the field-recommended concentrations (100 mg a.i. L^−1^) exceeded the LC_50_ by 830-fold, indicating high risk of population suppression. The no-observed-effect concentration (NOEC) concept, defined as the highest exposure level for which no (adverse) effects were observed can also be considered for risk assessment, especially about long-term studies [51]. The NOEC values of imidacloprid and hexaflumuron on the effects on reproduction and growth of *Coccinella septempunctata* were found to be 3.42 g a.i. ha^−1^ and 1.52 g a.i. ha^−1^, respectively [52]. The NOEC for *Trichogramma brassicae* exposed to thiamethoxam was 0.05 μg a.i. L^−1^ [53], while for *T. evanescens*, it was 1.2 μg a.i. L^−1^ [54].

Concentrations below HC_5_ could align with IPM if sublethal effects are minimized. Therefore, it is necessary to evaluate the compound’s sublethal effects of other species to find the NOEC values. Because there are few tests on NOEC in the database, we only used acute toxicity data for sensitivity analysis. In the present study, we evaluated the effects of the pesticide acetamiprid on *T. dendrolimi* using four toxicity biological parameters: parasitism percentage, emergence percentage, survival rate, and development period. Our 21-day chronic exposure study established a NOEC of 2.3 μg a.i. L^−1^ for both parasitism and emergence rates in *T. dendrolimi*, representing an important addition to the limited existing database on parasitoid NOECs. Sublethal exposure effects manifested primarily as reduced parasitism rates (≥2.3 μg a.i. L^−1^) and altered developmental dynamics (11.5–57.5 μg a.i. L^−1^), despite no adverse effects on survival rates within 14 days post-emergence. This pattern of sublethal impacts, including reduced reproductive capacity without affecting survival, has been documented across multiple parasitoid systems [55,56,57,58], suggesting common mechanisms of action. Sublethal effects on parasitoids include reduced fecundity in *T. brassicae* [52] and altered host-searching behavior in *Aphidius ervi* [59]. The observed reduction in parasitism and emergence rates without effects on developmental duration or offspring survival suggests specific sublethal mechanisms of action. Behavioral disruption appears predominant, with acetamiprid’s known neurotoxic effects on hymenopteran chemoreceptions [60], potentially impairing host location and oviposition activities. This aligns with findings in *T. chilonis* where neonicotinoids reduced foraging efficiency by 40–60% [48]. At the physiological level, acetamiprid’s action as a nicotinic acetylcholine receptor antagonist [19] may specifically affect adult nervous system functions required for parasitism, while sparing developmental processes in offspring. This explains the dichotomy between reduced parasitism rates but unaffected developmental duration, a pattern also observed in *T. brassicae* exposed to imidacloprid [61].

The ecological ramifications of these findings are profound. The SSD curves in Figure 5 allowed a comparison of the relative acute toxicity of *T. dendrolimi* to acetamiprid with other parasitic species reported in the open literature [36,37]. Notably, *T. dendrolimi* exhibited greater susceptibility to acetamiprid than six commonly deployed biocontrol agents: *G.* cf. *brasiliensis*, *T. nubilale*, *T. evanescens*, *T. japonicum*, *T. ostriniae,* and *T. confusum*. These interspecific comparisons employed standardized exposure protocols (equivalent routes and durations), ensuring methodological consistency and data reliability. This pronounced sensitivity gradient among parasitoid species carries significant implications for integrated pest management (IPM) systems where multiple wasp species provide complementary pest control services [62]. Quantitative risk assessment revealed alarming exposure scenarios: field-recommended concentrations (100 mg a.i. L^−1^), our SSD analysis predicts 76.8–97.9% population impacts, which could severely compromise biological control services. The HC_5_ value (0.11 mg a.i. L^−1^) was 48-fold higher than the NOEC (2.3 μg a.i. L^−1^), which underscores the importance of sublethal assessments, as standard acute toxicity tests may underestimate ecological risks [55]. A compound that kills 50% of the beneficial arthropods can be more acceptable than the one that decreases their fecundity and parasitism potential and makes the surviving individuals malformed. This challenges current risk assessment paradigms—while IOBC classifies insecticides with LC_50_ < 0.01 mg a.i. L^−1^ as ‘harmful’ [63], our SSD analysis suggests HC_5_ (0.11 mg a.i. L^−1^) may better predict field-level risk.

Formulation-dependent toxicity represents another critical consideration. Comparative studies demonstrate commercial acetamiprid formulations as exhibiting enhanced toxicity relative to pure compounds at equivalent active ingredient concentrations [64,65]. Commercial products exhibit higher acute toxicity to Hymenopteran species at equivalent active ingredient concentrations, with surfactant-enhanced bioavailability identified as a primary mechanism [64,65]. The adjuvants in these formulations induce prolonged sublethal behavioral alterations, including >48 h locomotor inhibition and 40–60% reduction in host-seeking efficiency [64]. These effects correlate with observations where inert ingredients in agricultural formulations amplify pharmacological impacts through improved cuticular penetration [64,65]. Field-realistic concentrations (1.25 g·L^−1^) significantly impair reproductive functions, with 25–35% egg-laying reduction persisting at 0.1× field doses [64,66]. The degradation dynamics of commercial formulations in soil systems further complicate risk assessments, as organic matter content modulates their persistence [65]. These formulation effects necessitate urgent revision of regulatory protocols that currently evaluate only pure compounds, systematically underestimating ecological risks [64,66].

A limitation of our study was that we only used an inertial substrate glass tube to establish the exposure–response relationships of acetamiprid on *T. dendrolimi*. Future studies should investigate the effects of acetamiprid under more realistic field conditions, considering factors such as sunlight, wind, and rain that may alter compound residues and toxicity. However, under the open-field condition, the compound residues are produced due to multiple degrading factors, such as sunlight, wind, and rain [67], and may not exert high toxicity on the parasitic wasps as described in this study. Therefore, further studies need to be conducted to identify the compounds that can be used safely under field scenarios without breaking the balance of the agroecosystem. In addition, in the sublethal toxicity test, the NOEC values have their drawbacks when used to evaluate the risk of pesticides. Because in actual conditions, the recommended maximum field application rates of pesticides will often be higher than the NOEC values. Evaluative indicators with more different values should be considered to assess the effects of the sublethal doses of pesticides. For example, 50% of the affected *Trichogramma* wasps are used to ensure the recovery of non-target arthropod populations. In addition, our sublethal tests focused solely on *T. dendrolimi*; future work should evaluate NOECs for other parasitoids.

## Figures and Tables

**Figure 1 insects-16-00698-f001:**
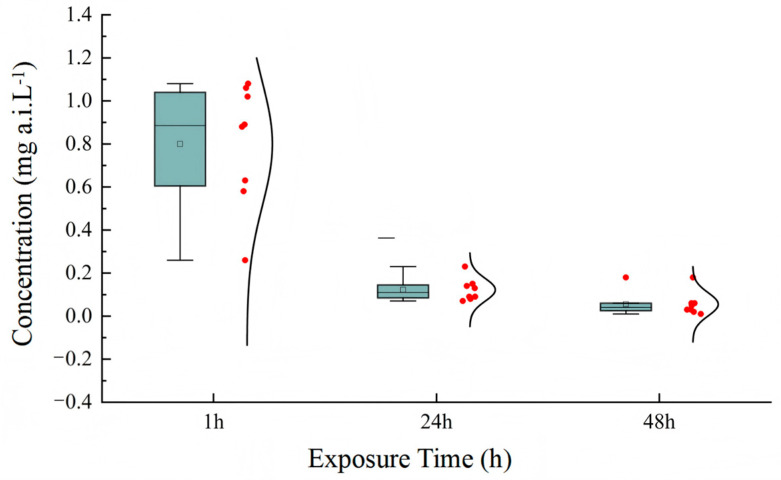
Box plots representing the LC_50_ mean values for acetamiprid as calculated for *T. dendrolimi* in the tests with the active ingredient. Eight independent tests (each with 7 concentrations of acetamiprid) were performed to determine LC_50_ values. Box plots show LC_50_ values at three exposure times (24 h, 48 h, 72 h; *n* = 8 tests total).

**Figure 2 insects-16-00698-f002:**
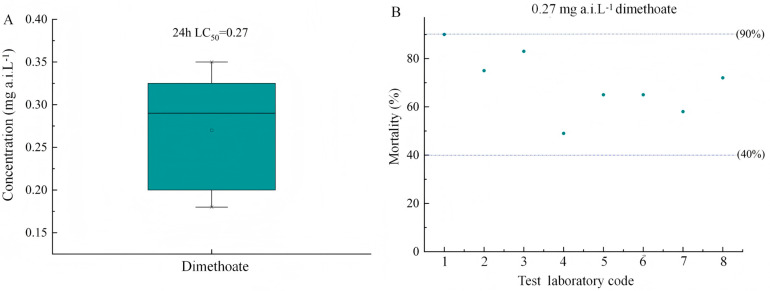
The toxicity of reference item dimethoate to *T. dendrolimi*: (**A**) the 24 h LC_50_ mean values of dimethoate after the egg parasitoid was exposed to dimethoate; (**B**) mean mortality (±SE) of *T. dendrolimi* adults exposed to dimethoate at 0.27 mg a.i. L^−1^ (*n* = 8 replicates). The number on the *x*-axis refers to eight different toxicity tests conducted on *Trichogramma dendrolimi* using dimethoate as a reference insecticide.

**Figure 3 insects-16-00698-f003:**
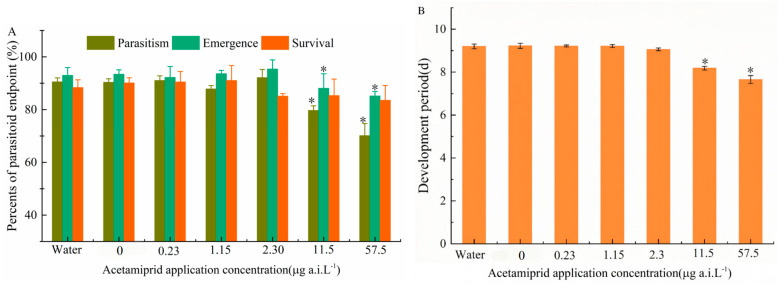
(**A**) The effect of different acetamiprid concentrations on the parasitism rate, emergence rate, and next-generation survival rates; (**B**) the development period of *T. dendrolimi*. The asterisks ‘*’ indicate significant differences between the treatment and control samples at *p* < 0.05 (ANOVA, Tukey’s HSD).

**Figure 4 insects-16-00698-f004:**
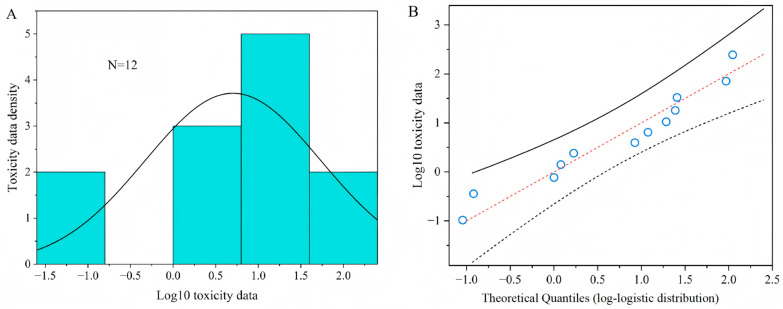
Distribution and goodness of fit test of toxicity data. (**A**) A histogram and fitted density plot of toxicity data of acetamiprid; (**B**) a Q-Q plot of toxicity data versus fitted log-logistic distribution of acetamiprid.

**Figure 5 insects-16-00698-f005:**
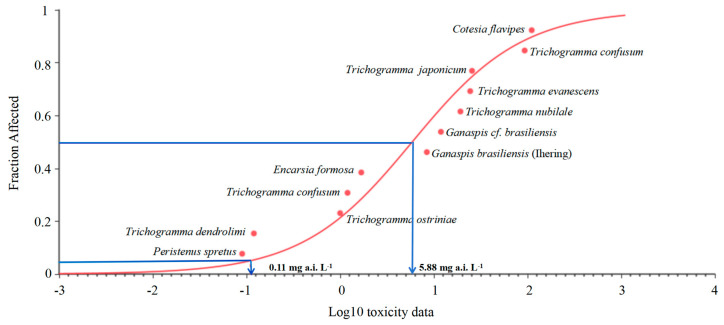
Species sensitivity distribution (SSD) constructed based on LC_50_ values for acetamiprid obtained in this study for *T. dendrolimi* and from literature for other parasitoid.

**Table 1 insects-16-00698-t001:** HC_5_, HC_50_, and PAF values of acetamiprid to parasitic species.

Pesticide	Field-Recommended Concentration (mg a.i. L^−1^)	HC_5_ (mg a.i. L^−1^)	HC_50_ (mg a.i. L^−1^)	PAF
acetamiprid	30~100	0.11	5.88	76.8~97.9%

## Data Availability

The raw data supporting the conclusions of this article will be made available by the authors on request.

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
