# Peer review of "Acetamiprid-Induced Toxicity Thresholds and Population Sensitivity in Trichogramma dendrolimi: Implications for Pesticide Risk Assessment"

_insects, 2025, doi:10.3390/insects16070698_

Round 1
Reviewer 1 Report
Comments and Suggestions for Authors
This study provides a systematic investigation into the acute and sublethal toxicity of the neonicotinoid insecticide acetamiprid on Trichogramma dendrolimi, a key biological control agent. The authors innovatively applied a species sensitivity distribution (SSD) model to benchmark T. dendrolimi's sensitivity against other parasitoid species, offering a novel comparative framework between the hazardous concentration for 5% of species (HC5) and the no-observed-effect concentration (NOEC). The study addresses ecotoxicological risks of acetamiprid with valuable insights for pesticide regulation. The manuscript is well written, the background is detailed, and the conclusions are supported by the results. However, some revisions to strengthen experimental rigor are essential. Post-revision, this work will significantly advance IPM strategies and risk assessment frameworks.
Specific Comments:
- The rationale for selecting specific sublethal concentrations (e.g., 1/2, 1/5 LCâ‚…â‚€) is unclear. Recommendation: Provide scientific justification (e.g., preliminary experiments or literature support) for these fractions in the Methods section.
- Sublethal effects show reduced parasitism and emergence rates but no impact on developmental duration or offspring survival. Recommendation: Expand the Discussion to explore mechanisms (e.g., behavioral inhibition, physiological damage) and ecological implications, supported by comparative literature.
- Table 1 does not clarify how PAF values were derived (e.g., SSD extrapolation or field data). Recommendation: Describe calculation methods or cite formulas in the Methods.
- Older references dominate. Recommendation: Update citations to include recent studies (2022–2025) on parasitoid-pesticide interactions to reflect current advancements.
- The Discussion lacks comparisons with other neonicotinoids (e.g., clothianidin, dinotefuran). Recommendation: Add a comparative toxicity analysis to emphasize the novelty of this work.
Reviewer 2 Report
Comments and Suggestions for Authors
The value of the results is limited for practical use, as only the pure active ingredient has been investigated, Th inclusion of one or two commercial plant protection products would have improved the usability of the results for agricultural practice.
Statistical methodology: is a bit dispersed in the txt
sector 2.4 is also related to statistical methodology. Tox data are described correctly as log-distributed. In figures 1 and 2 this is ignored, and concentrations are displayed on a linear scale.
typographical errors:
Line 179 NOEC
Line 312:
" The HC5 value appears to be approximately 456 times than NOEC..."
must be wrong.
HC5= 110 µg a.i. L-1
NOEC=2.3 µg a.i. L-1
Quotient~48
discussion should include some statements to strength and weakness of this study.
Reviewer 3 Report
Comments and Suggestions for Authors
This manuscript addresses an important issue in ecotoxicology and integrated pest management, evaluating the acute and sublethal toxicity of acetamiprid to Trichogramma dendrolimi, a key parasitoid used in biological control. The authors also apply species sensitivity distribution (SSD) analysis to assess ecological risk at the population level. The study is timely and relevant, and the experimental approach is generally appropriate. However, several issues related to clarity, data presentation, and interpretation need to be addressed before the manuscript can be considered for publication.
- Abstract
The abstract is intended to briefly present the theme and scope of the study, providing a concise overview of the research context and objectives. However, the authors begin the abstract by immediately discussing their results without offering any foundational background or rationale for the study.
Recommendation: I strongly suggest rewriting the abstract to include:- A brief context and statement of the problem.
- The significance of studying acetamiprid’s effects on dendrolimi.
- A succinct description of methods used.
- Key findings and their implications.
- Figure 1 – Data Redundancy and Axis Label
The distribution plot (box plot) in Figure 1 effectively shows that the LC50 values decrease with increased exposure time. Adding a separate curve appears redundant unless it offers distinct or supplementary information.
Recommendation:- Clarify the purpose and added value of the curve alongside the box plot.
- Clearly label the x-axis to indicate whether it represents exposure time, replicate sets, or test conditions.
- Figure 2b – Unclear Axis and Explanation
Figure 2b is not adequately explained in the text. The x-axis labeled as "Test laboratory code" is ambiguous.
Recommendation:- Provide a clear explanation of what each code or bar on the x-axis represents (e.g., different laboratory strains, replicates, or geographic origins).
- Justify the relevance of presenting this data format.
- Dimethoate Time-Dependent Data Missing
The authors presented time-dependent data for acetamiprid but did not do the same for the reference compound dimethoate.
Recommendation:- Explain why a time-dependent experiment was not conducted for dimethoate.
- If feasible, include such an experiment or discuss limitations and justify its exclusion.
- Line 93 – Incomplete Sentence
The sentence beginning on line 93 reads:
"To investigate the long-term sublethal effects of acetamiprid exposure on key biological of T. dendrolimi,"
This is grammatically incomplete and unclear.
- Figure 4B – X-Axis Labeling
In Figure 4B (Q-Q plot), the x-axis labeling is missing or unclear.
Reviewer 4 Report
Comments and Suggestions for Authors
The manuscript by Zhang et al. assesses the acute and sublethal effects of the insecticide acetamiprid on the egg parasitoid Trichogramma dendrolimi. The results indicate a very low LC50 (0.12 mg active ingredient per liter), significantly below typical field application rates. Sublethal doses were found to significantly reduce parasitism and emergence. Sensitivity analysis further reveals that Trichogramma dendrolimi is highly vulnerable, with field application rates potentially affecting up to 97.9 percent of the population. I have several major concerns regarding the manuscript, along with additional suggestions outlined below. In its current form, the manuscript is not acceptable due to serious methodological, structural, and citation-related issues. Moreover, the study hypothesis does not align with internationally accepted standards. For example, concentrations below LC30 are generally considered safe and compatible with integrated pest management (IPM) practices.
Major Concerns:
- The manuscript lacks a clear structure. Dimethoate is not mentioned in the abstract and is barely discussed in the results section.
- There are major concerns with the methods used. The presentation of lethal concentration data is inadequate. The bioassays used to assess sublethal effects are not widely accepted in ecotoxicology. Furthermore, replication numbers are too low—only five replicates per treatment for sublethal effects and just three for lethal concentration calculations.
- Several references are misattributed. Specific issues are noted below.
- Numerous terminological errors appear throughout the manuscript and need correction.
- Some parts of the manuscript are inconsistent with the methods described in the Materials and Methods section.
- Parameters such as NOEC are not clearly explained. The use of species sensitivity distribution (SSD) analysis is unusual for this type of ecotoxicological study and needs better justification.
Miscellaneous Comments and Suggestions:
L15: Clarify “three orders.” Do you mean threefold?
L16: Define NOEC clearly.
Introduction: Needs rewriting for clarity. The section includes several miscitations and lacks a thorough literature review.
L37: Abbas et al. (2025) is unrelated to Trichogramma; please replace with a relevant reference.
L47: In addition to Cheng et al. (2018), consider citing work by Zhang et al. and others on sublethal effects of pesticides on Trichogramma species.
L50–51: Include additional relevant references. For example: Chen X, Song M, Qi S, Wang C. Safety evaluation of eleven insecticides to Trichogramma nubilale (Hymenoptera: Trichogrammatidae). J Econ Entomol. 2013;106(1):136–41.
L61: Wang et al. (2025) pertains to Encarsia formosa, not egg parasitoids. This is a miscitation.
L76: The "Guidance 2016" document must be properly cited with full reference details.
L92: Clarify what is meant by “long-term assessment” of sublethal effects.
L102–103: This statement exceeds the study’s stated aims.
L113–114: Rephrase for clarity. Clarify the type of exposure—were fresh eggs exposed to 48-hour-old females for 24 hours?
L134: Specify the seven tested doses.
L154: Clarify whether Yu et al. (2014a) tested imidacloprid rather than acetamiprid.
L157: Five replications are insufficient for robust conclusions.
L160: Clarify methodology regarding the use of treated tubes and the number of eggs. Specify whether eggs were irradiated or cold-stored to stop development.
L166–167: The methodology is flawed—parasitized egg counts do not equal emergence due to common superparasitism in Trichogramma.
L168: Were hatched parasitoids individually transferred to clean tubes?
L170–172: These sentences repeat earlier content.
L172–173: Provide full details of data analysis—software, statistical tests, transformations used, etc.
L178: Specify which database was used, the keywords applied, and how many papers were found.
L197: Identify the eight tests. Lethal concentration analysis must include chi-square, slope, standard errors, and confidence intervals. Figure 1 lacks time-point details (e.g., 1h, 24h, 48h).
L208: Clarify whether eight doses or eight tests were conducted.
L210: Check for typos in the axis of Figure 2b. The figure lacks scientific clarity.
L210–212: It is unclear why dimethoate was included. No side-effect bioassays on Trichogramma dendrolimi are presented.
L217: Report full ANOVA details: F-values, degrees of freedom, and p-values.
L221: Explain how LOEC and NOEC values were calculated. These details are missing from the methods.
Figure 3a: The text refers to 0.23, not 2.3 mg a.i./L—please correct.
L227: Authors mention Tukey’s test but report LSD test—this inconsistency must be resolved.
L272: LC95 is not reported anywhere—please include or remove reference to it.
L278: What is NOER? Define it clearly.
L280–281: Cite papers relevant to parasitoids, not cladocerans.
L284: Replace "endpoints" with "biological parameters."
L288: Clarify what pesticides or doses are meant.
L298: The assumption that LC50 is incompatible with IPM contradicts IOBC standards. Concentrations below LC30 are generally acceptable.
L301: He et al. and Gholamzadeh et al. (2014) focus on predators. Please cite relevant studies on parasitoids.
L312: Is it 456 times or 456-fold higher?
L317: Specify that the species studied is Trichogramma dendrolimi only.
L327–328: See related comments above.
References:
L427: Clarify the identity of "P. poorjavad."
L429: Verify citation details for Raven and Nahrung (2020).
Comments on the Quality of English Language
Please refer my report for details.
Round 2
Reviewer 4 Report
Comments and Suggestions for Authors
I have quickly reviewed the revised version of the manuscript titled "Acetamiprid-Induced Toxicity Thresholds and Population Sensitivity in Trichogramma dendrolimi: Implications for Pesticide Risk Assessment" as well as the corresponding response letter to the comments. Below are my comments and suggestions for improving the manuscript:
The reference to Baysal et al., 2021 (Line 350) appears to be incorrect. This paper is unrelated to parasitoids, focusing instead on colorectal adenocarcinoma. I recommend that the authors carefully review the list of references cited in the manuscript. Any irrelevant or misquoted citations should be removed or replaced with appropriate references that are relevant to the topic at hand.
On line 332, the species name G. cf. brasilensis is incorrectly capitalized. As per standard taxonomic conventions, the genus name should be capitalized, but the species name should be in lowercase. I recommend correcting the formatting to ensure it adheres to proper scientific naming conventions.
While the manuscript discusses sublethal effects on Trichogramma, the authors also cite studies related to other parasitoid groups, such as aphid parasitoids. Because the manuscript’s focus on Trichogramma dendrolimi, I suggest the authors include more references specifically related to sublethal effects in Trichogramma species, rather than citing studies on other parasitoid groups. This will help maintain the manuscript’s focus and strengthen its relevance to the target species.
In addition to the above issues, the manuscript would benefit from a thorough editing to address various language and formatting inconsistencies. A careful review will enhance the clarity and readability of the paper, making it more polished for publication.
Please note that this review was conducted with a quick look at the manuscript, and I believe further detailed revisions may enhance the quality of the paper.
I hope these suggestions help improve the manuscript. I look forward to seeing the detailed revised version.
Sincerely,
Comments on the Quality of English Language
I recommend that text of the manuscript be proofread by a native English speaker.
